# Circular RNA—Is the Circle Perfect?

**DOI:** 10.3390/biom11121755

**Published:** 2021-11-24

**Authors:** Lavinia Caba, Laura Florea, Cristina Gug, Daniela Cristina Dimitriu, Eusebiu Vlad Gorduza

**Affiliations:** 1Department of Medical Genetics, Faculty of Medicine, “Grigore T. Popa” University of Medicine and Pharmacy, 700115 Iasi, Romania; vgord@mail.com; 2Department of Nephrology-Internal Medicine, Faculty of Medicine, “Grigore T. Popa” University of Medicine and Pharmacy, 700115 Iasi, Romania; lflorea68@yahoo.com; 3Microscopic Morphology Department, “Victor Babes” University of Medicine and Pharmacy, 300041 Timișoara, Romania; dr.cristina.gug@gmail.com; 4Department of Morpho-Functional Sciences II, Faculty of Medicine, “Grigore T. Popa” University of Medicine and Pharmacy, 700115 Iasi, Romania; daniela.dimitriu@umfiasi.ro

**Keywords:** circular RNA, biomarker, cancer

## Abstract

Circular RNA (circRNA) is a distinct class of non-coding RNA produced, in principle, using a back-splicing mechanism, conserved during evolution, with increased stability and a tissue-dependent expression. Circular RNA represents a functional molecule with roles in the regulation of transcription and splicing, microRNA sponge, and the modulation of protein–protein interaction. CircRNAs are involved in essential processes of life such as apoptosis, cell cycle, and proliferation. Due to the regulatory role (upregulation/downregulation) in pathogenic mechanisms of some diseases (including cancer), its potential roles as a biomarker or therapeutic target in these diseases were studied. This review focuses on the importance of circular RNA in cancer.

## 1. Introduction

Circular RNAs (circRNAs) are endogenous RNA molecules that form covalently closed continuous loops [1,2]. They are part of the long non-coding RNA (lncRNA). LncRNA can be classified into the following several categories: based on transcript length, association with annotated protein-coding genes, association with other DNA elements of a known function, protein-coding RNA resemblance, association with repeat elements, association with specific biochemical pathways, implication in biochemical stability, conservation of the sequence and structure of cellular elements, differential expression in different tissues, association with subcellular structures, and variable function [3].

CircRNAs are not only specific to humans. It has also been discovered in others Metazoa (mouse, Zebrafish, Caenorhabditis elegans, Saccharamyces cerevisiae, Schizosaccararomyces pombe, Plasmodium falciparum, Dictyostelium discoideum) or vegetals (Arabidopsis thaliana Oryza sativa ssp. Japonica, Oryza sativa ssp. Indica, Nicotiana benthamiana etc) [4].

CircRNA molecules, depending on their composition, can be classified into the following several categories: exonic circRNAs (ecircRNAs)—formed by exons, circular intronic RNAs (ciRNAs)—formed by introns, and exon-intron circRNAs (EIciRNAs)—formed by both exons and introns [5]. Most circRNA molecules contain exons from genes encoding proteins, but they also originate from introns, intergenic regions, UTRs, ncRNA loci, and antisense locations of known transcripts. The following other newly described categories have been added: fusion circRNA (f-circRNAs), read-through circRNAs (rt-circRNAs), and mitochondria-encoded circRNA (mecciRNAs). In this review, we summarise data on the classification, biogenesis, and functions of circRNAs and discuss their potential role as biomarkers, especially in the diagnosis of cancers.

## 2. Circular RNA—Complex Molecule

### 2.1. Prevalence

Over 70% of the human genome gives rise to different RNA molecules with a role in maintaining tissue homeostasis and pathophysiological processes [6]. Only 2% of all RNA molecules encode proteins [7]. Over 10% of expressed genes can produce circRNAs [8]. Of the active transcribed genes in humans, 5.8–23% of them give rise to circRNAs [9]. A circRNA is expressed in a tissue-specific complex way, cell-type specific or developmental-dependent stage [10,11,12,13]. They are found with increased frequency in the brain and during foetal development [14]. Nearly 20% of protein-coding genes produce circRNAs in mammalian brains [15]. In the most comprehensible database-circAtlas, a number of the 413,657 types of circRNAs were described in 2020. By categories in descending order, they were as follows: exonic—252,494; intronic—51,291; antisense—39,060; non-repeat—24,896; 3′UTR—1686; 5′UTR—754 [16]. Salzman et al. estimate that the circRNAs represent 1% of (polyA)RNA molecules [17].

The newly described entity, mecciRNAs, represents only a small part of the total circRNA molecules [18]. Hundreds of circRNAs encoded by the mitochondrial genome have already been identified [19].

### 2.2. Properties of circRNA

CircRNA is resistant to RNaseR (an enzyme with 3′ to 5′ exonuclease activity), which degrades most linear RNAs [20,21]. The half-life is 2.5 times longer than that of linear mRNA, which gives them characteristics used as diagnostic or therapeutic biomarkers [22,23]. They have no electric polarity and are not a polyadenylated tail [24]. The length varies between 100 and 4000 nucleotides [12].

### 2.3. CircRNAs Localisation

Different types of circRNA are located in different cell compartments. ciRNAs are found in the nucleus, ecircRNAs are present in the cytoplasm and exosomes, EIciRNA is located in the nucleus, f-circRNAs are present in all cellular compartments, rt-circRNAs are found in cytoplasm, while mecciRNAs are located in the mitochondrial environment (inside and outside of organite) [25,26].

Stable circRNAs were discovered in exosomes with a role in the transcriptional and translational regulation, splicing regulation, miRNA sponge, and protein inhibition [27]. They may have an oncogenic role or a tumour suppressor one [28], being involved in tumour growth and metastasis, angiogenesis, modulation of the microenvironment, pre-metastatic niche formation, and immunomodulation and drug resistance [29]. Some exosomes survive in several biological products such as blood, urine, saliva, breast milk, synovial fluid, amniotic fluid, bronchoalveolar lavage fluid, malignant ascites, and semen [27,30]. More than 1000 exosomal circRNAs have been identified in human serum and allowed a differentiation between tumour patients and healthy ones [27].

### 2.4. Biogenesis

Most circRNA molecules contain several exons, often two or three exons [13]. Since the vast majority of exonic circRNAs contain exons placed in the middle of the reading frame, it has been suggested that the mechanism of formation of circRNAs is mainly related to RNA splicing [13]. Alternative circularisation is the phenomenon by which multiple exon circularisation events occur from single loci genes [13]. The intronic repeats can be a determining factor of the back-splicing process [31,32]. Approximately 90% of circRNAs appear to have complementary ALU elements in introns that flank the gene region from which they originate [31,33].

For exonic circular RNAs, the common mechanism is back-splicing in which 3′ splice donors bind covalently to 5′ splice acceptors but in reverse order. There are the following three models: exon skipping or lariat-driven circularization; direct back-splicing, or intron-pairing driven circularization and RNA-binding-protein-driven circularization (Figure 1) [23,34].

Additionally, three mechanisms have been proposed in formation of intronic circRNA. In the first model the stages are the isolation of an exon, the assembly of the isolated exon with the adjacent one and the formation, in this way, of a linear intron, followed by its circularisation and finally the loss of the 3′-tail of the lariat RNA. In the second model, the second exon is isolated (downstream exon), and from the upstream exon coupled with the downstream intron, the exon is detached, and the lariat is circulated. In the third spliceosome model, the 3′end of the intron is connected to the 5′ end and forms the lariat from which the 3′-tail will be removed [34].

The back-splicing efficiency seems to be lower than canonical splicing [24,35]. For circularisation, the presence of canonical splice signals is required [35]. The formation of circRNA by back-splicing is correlated with a minimal exon length of 353 nucleotides (nt) in the case of a single exon circularised by back-splicing and 112–130 nt per exon in the case of multi-exon back-splicing [13,24]. In the majority of cases, exons that will be circularised are flanked by introns that contain reverse complementary sequences. These can be repetitive elements (ex-ALU elements) or non-repetitive complementary sequences. The number of nucleotides in these sequences is 30–40, but the longer the sequence is, the higher the circRNA production is [13,24].

Circular RNAs are more common than linear transcripts in about 50 human genes [17]. Factors influencing circRNA formation are the intron and exon length, repetitive sequences and RNA-binding proteins (RBPs) such as quaking (QKI), RNA-specific adenosine deaminase (ADAR1), NF90/NF110, heterogenous ribonucleoprotein L (HNRNPL), and muscle blind (MBL/MBNL1) [25,36,37].

Most circRNAs are synthesized in a sense orientation, but there are also molecules of circRNAs synthesized in an antisense orientation [38].

Jeck et al. (2013) show that circularised exons are flanked by introns that have ALU repeats (twice as common as uncircularised exons) [39]. These ALU elements are in inverted orientation rather than complementary. The proportion of introns that flank ecircRNA and have complementary ALU pairs and those that have non-complementary pairs is 20 and 8%, respectively [39]. Circularised exons are six times more likely to contain complementary ALUs than noncircularised exons, and ecircRNAs may be 10 times (> 10-fold) more common than associated linear transcripts [39]. Such a particularity was identified in the central nervous system [32].

F-circRNAs are formed by a linear fusion transcript derived from genome rearrangements, while rt-circRNAs are formed by the coding exons of two adjacent genes with similar orientations (read-through transcriptions) [5,25,40,41]. Over 95% of human genes are alternately spliced, a process regulated by both cis-regulatory elements and trans-acting factors [42]. Serine and arginine-rich (SR) proteins are RNA-binding proteins that act as regulators of constitutive and alternative splicing [43]. Heterogeneous nuclear ribonucleoproteins (hnRNPs) represent key proteins in the cellular nucleic acid metabolism intervening in alternative splicing, mRNA stabilisation, and transcriptional and translational regulation [44].

### 2.5. CircRNA Functions

CircRNA functions are manifested at the molecular level, the cellular level, and the organism level, intervening in both normal physiological processes and in pathological processes. CircRNAs modulate the expression of hereditary information by intervening in the following molecular mechanisms of gene expression: transcription, splicing, microRNA sponge, modulation of protein–protein interaction and protein (RBP) sponges, as scaffolds for the assembly of the components’ translation (Figure 2) [45]. The following two alternative mechanisms have been proposed for initiating translation: the presence of an Internal Ribosome Entry Site (IRES) and the presence of N6-methyladenosine (m6A) residues, both pathways being activated under cellular stress conditions [46]. Proteins encoded by circRNAs are truncated proteins with the same function as full-length proteins, others have independent or opposite functions [23].

#### 2.5.1. m6A Modifications

At least 13% of all circRNAs have m6A modifications and a single m6A is sufficient to trigger translation in a cap-independent manner [46,47]. An m6A modification regulates the circRNA metabolism precisely by modulating biogenesis, translation, degradation, and cellular localization [48]. m6A modification appears to increase the biogenesis of circRNAs in a METTL3/YTHDC1 (methyltransferase-like 3 protein/YTH domain containing 1)-dependent manner [48,49]. In the human heart, 9% of expressed genes can produce circRNAs, but, in the brain, the percentage is higher: 20% [48,50]. m6A sites in mRNAs are more common in the last exon, but the last exon does not enter in the circularisation process, which suggests that the pattern of m6A modification in mRNAs and circRNAs is different [13,48]. The functions of m6A on mRNA and on circRNAs are mediated by the following three categories of factors: methyltransferase (acts as “writers”), demethylase (acts as “erasers”), and factors involved in recognition (acts as “readers”) [48].

#### 2.5.2. CircRNA–Proteins Interactions

In a hypothetic situation of two proteins, A and B, which interact with each other, the interaction of circRNA with them can be in one of the following ways:It binds to both proteins and strengthens the interaction between them (cements). This effect is achieved by the following two mechanisms: circRNA mediates the post-translational changes (ubiquitination and phosphorylation) of protein A catalysed by protein B or the transactivation of protein A by protein B followed by subsequent changes;It binds only to one of the proteins that strengthen or dissociate the interaction between the two proteins;It binds to both proteins and dissociates them (normally they combine) [23].

#### 2.5.3. Binding or Sequestration of Proteins

The main functions of circular RNA are the sequestration of microRNA/proteins, the modulation of transcription-modulating RNA polymerase II (Pol II) transcription, interference with splicing, and translation to produce polypeptides [51].

CircRNAs bind to cis elements and control TFs (transcription factors) and modulate epigenome. There are the following three categories of actions: recruiting TFs, recruiting modifying enzymes, and recruiting chromatin remodelers [23]. Intronic circRNA (especially present in the nucleus) may be a positive regulator of RNA polymerase II (Pol II) transcription. EIciRNAs (also located in the nucleus) interact with U1 small nuclear ribonucleoprotein particles (U1snRNP) and thereby promote the transcription of parental genes [52].

Exonic circRNA reaches the cytoplasm in the following two ways: by a nuclear export system or by escaping from nuclei during cell division [53,54]. Different factors intervene in the nuclear export depending on the size of the circRNA: UAP56 is involved in the export of circular RNAs’ molecules with a length > 1300 nt and URH49 in the nuclear export of short circular RNAs (<400 nt) [23]. UAP56 is an ATP-dependent helicase that is an essential splicing factor also important for mRNA export [55]. URH49 (which shares a sequences identity of 90% and similarity of 96% with UAP56) have an increased level of expression in the cell proliferation phase. Thus, it appears that a URH49-dependent mRNA export is related to cell proliferation [56].

The regulation of circRNA production is performed by cis and trans elements. The cis elements are represented by intronic complementary sequences (ICSs), and the trans elements are represented by RNA binding proteins (RBPs). CircRNA in the cytoplasm acts mainly by binding and trapping microRNAs. CircRNAs in the nucleus regulate expression and splicing in host genes [6,11,57]. circRNAs have specific binding sites for miRNAs and, thus, act as a “trap” for miRNAs that no longer interfere with mRNA expression [58].

Intronic and exon-intron circRNAs act as regulatory factors for parental gene transcription through interactions with the RNA polymerase II, snRNPs, and hnRNPs [46,52].

### 2.6. Circular RNA in Essential Processes—Cell Cycle, Proliferation, Apoptosis

Circular RNAs are involved in the following key cellular processes: proliferation, apoptosis, cell cycle [12]. They can intervene in the cell cycle: circ-Foxo3 interacts with CDK2 (cyclin-dependent kinase 2) and cyclin-dependent kinase inhibitor 1 (p21) and this circ-Foxo3-p21-CDK2 complex blocks cell cycle progression and represses all proliferation [59].

Intronic circRNAs can accumulate in the cytoplasm and bind to TAR DNA binding protein 43 (TDP43) and this could be a beneficial effect in the treatment of amyotrophic lateral sclerosis because it suppresses TDP43 toxicity [60].

## 3. Circular RNA—A Potential Biomarker in Cancer

Biomarkers can be classified, according to Food and Drug Administration and the National Institutes of Health, into the following several categories: susceptibility/risk; diagnostic; monitoring; prognostic/predictive; pharmacodynamics/response biomarker; safety biomarker, but some biomarkers can fit it into several classes of biomarkers [61].

CircRNAs are involved in the hallmarks of cancer-sustaining proliferative signalling, the evasion of growth suppressors and/or the impairment of differentiation signals, avoiding immune destruction, enabling replicative immortality, tumour-promoting inflammation, activating invasion, metastasis and angiogenesis, genome instability and mutation, evading cell death and senescence, deregulating cellular energetics, and therapeutic resistance in human cancers [62,63]. In some cancers, several circRNA molecules have been analysed, some with upregulation effects, others with downregulation effects. Table 1 presents statistics of the number and effect of circRNAs registered and characterised in the CircAtlas database as being associated with some pathologies.

For diagnostic accuracy, a biomarker should have an AUC (Area Under the Curve) greater than 0.5 on ROC (receiver operating characteristic) curves.

The potential circRNA diagnostic biomarkers in cancers are summarized in Table 2.

Some of the cRNA molecules have been shown to be more effective biomarkers than conventional tumour serum markers. The use of a combination of circRNAs or circRNAs and conventional tumour serum markers is also more efficient in some cancers or to discriminate between cancer and other pathologies.

In gastric cancer, the combined use of hsa_circ_0007507, CEA, and CA199 has the best diagnostic accuracy (AUC = 0.849) of distinguishing between gastric and healthy cancer patients. The use of only the serum tumour marker Carcinoembryonic antigen (CEA) has an AUC = 0.765, and the use of Carbohydrate antigen 19–9 (CA199) has a lower efficacy [77].

In hepatocellular carcinoma, the exo_circ_0006602 was a more efficient diagnostic biomarker than the classic tumour serum markers: alpha-fetoprotein (AFP) (AUC = 0.694, *p* < 0.002) and CEA (AUC = 0.589, *p* = 0.146). The combined use of exo_circ_0006602 and AFP increases the AUC to 0.942 (*p* < 0.00011) [83].

CircSMARCA5 plays a role in apoptosis, proliferation, invasion, and metastasis and has been shown to be used in the diagnosis of hepatocellular carcinoma with good accuracy: AUC = 0.938. It is also effective (alone or in combination with AFP) in differentiating HCC vs. Hepatitis and HCC vs. Cirrhosis. Thus, AUC values of 0.853 and 0.903 were obtained for circSMARCA5 and circSMARCA5 + AFP, respectively, in the discrimination of HCC vs. Hepatitis. For the differentiation of HCC vs. Cirrhosis, the use of circSMARCA5 brought an AUC of 0.711, and this value increased to 0.858 when combined with AFP. An important aspect is that in patients with AFP levels below 200 ng/mL, the diagnostic efficiency was good because the use of circSMARCA5 achieved an AUC of 0.847 in discriminating HCC vs. Hepatitis patients and an AUC of 0.706 in the HCC vs. Cirrhosis group. This is important because in 15–30% of patients with advanced HCC, the alpha-fetoprotein remains within the normal limits [84].

Cheng et al., in their review, summarised the following prognostic factors in bladder cancer: circLPAR1, circASXL1, circRIP2, circPICALM, circ_403658, circHIPK3, hsa_circ_0077837, and circEHBP1 [66].

In colorectal cancer, early diagnosis is very important because 20% of cases at the time of diagnosis already have metastases. The use of a panel of three circRNAs (circ-CCDC66, circ-ABCC1, circ-STIL) has a diagnostic accuracy expressed by an AUC of 0.78, and the association of carcinoembryonic antigen (CEA) and carbohydrate antigen 19–9 (CA19–9) markers increases the AUC at 0.855 [73,96].

CircRNAs with potential therapeutic targets are those involved in activating proliferation, growth, invasion, migration, and metastasis and negatively in apoptosis. This includes circRNAs with elevated levels [73]. Vice versa, low-level circRNAs are those that promote apoptosis and have a negative effect on growth, proliferation, migration, and metastasis. Most of these molecules act by interacting with miRNA molecules. The first category includes CDR1as/CiRS-7 (via miR-7), circHIPK3/circ_0000284 (via miR-7), circRNA-ACAP2 (via miR-21-5p), circ_000984 (via miR-106b), circ_0001955 (via miR-145), circ_0055625 (via miR-106b), circ-BANP, circ_0000826, circPPP1R12A/circ_0000423, circ_0001178 (via miR-382/587/616), circ-NSD2 (via miR-199b-5p), circ_001569 (miRNA-145), and circNSUN2 [73].

The second category includes circ-ITGA7 (via miR-370-3p), circDDX17, circ_0026344 (via miR-21, miR-31), circ-FBXW7, and circITCH (via miR-20a, miR-7, and miR-214) [73].

hsa_circ_0001946 also seems to be a potential biomarker prognosis for oesophageal squamous cell cancer (ESCC). Hsa_circ_0001946 intervenes in ESCC oncogenesis through its role in the miRNA–mRNA network and, thus, in cell proliferation, migration, and invasion [76].

Sand et al. found 23 circRNAs with significantly elevated levels in basal cell carcinoma and 48 circRNAs with low levels compared to healthy subjects. Different molecules of circRNAs come from the same *CASC15* gene (previous symbol *LINC00340*) and have different growth levels, ranging from 5.95 to 3.14 compared to levels in healthy subjects. Very low levels were recorded for the circRNAs derived from *FADS2* (has_circ_0022383 with −54.3; has_circ_0022392 with −41.36) or *ASAP2* (fold change −9.82) [97].

## 4. Conclusions

CircRNAs are abundant, stable, expressed in a cell-type and tissue-specific manner, and are present in accessible tissues, which makes them useful as biomarkers in cancer and other diseases. As these molecules are positively or negatively correlated with different stages and mechanisms in cancer, they could be used in diagnosis, risk, and prognostic stratification or therapeutic targets (e.g., via microRNA). Therefore, the use of circRNAs as biomarkers is another step towards personalised medicine. Further studies are necessary to validate some of these circRNAs as different types of biomarkers.

## Figures and Tables

**Figure 1 biomolecules-11-01755-f001:**
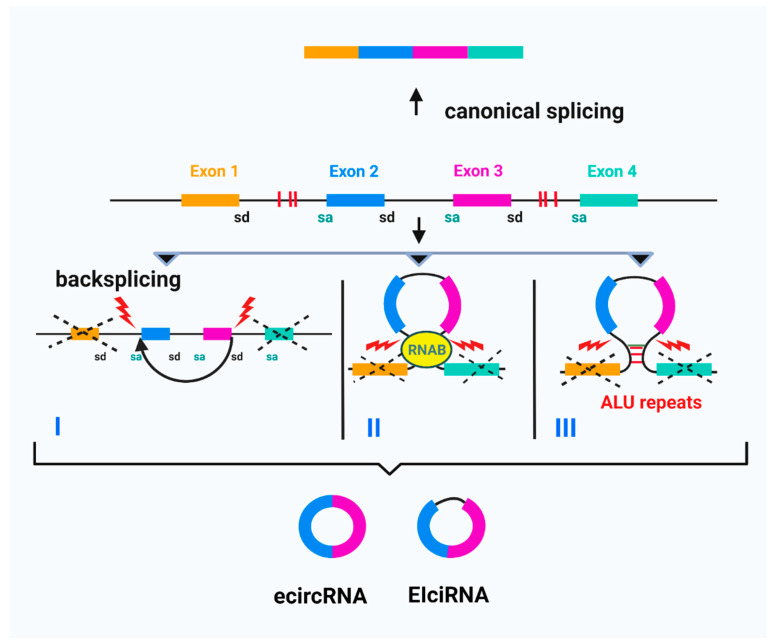
Biogenesis of ecircRNAs and EIciRNAs. I: Back-splicing—the splice donor binds covalently to splice acceptor; II: RNA-binding-protein-driven circularisation; III: Intron-pairing driven circularisation based on intronic ALU repeats. Adapted from “circRNA in Cancer”, by BioRender.com (2021). Retrieved from: https://app.biorender.com/biorender-templates (accessed on 18 November 2021). (sd—splice donor; sa—splice acceptor; RNAB—RNA binding protein; red bars—ALU repeats).

**Figure 2 biomolecules-11-01755-f002:**
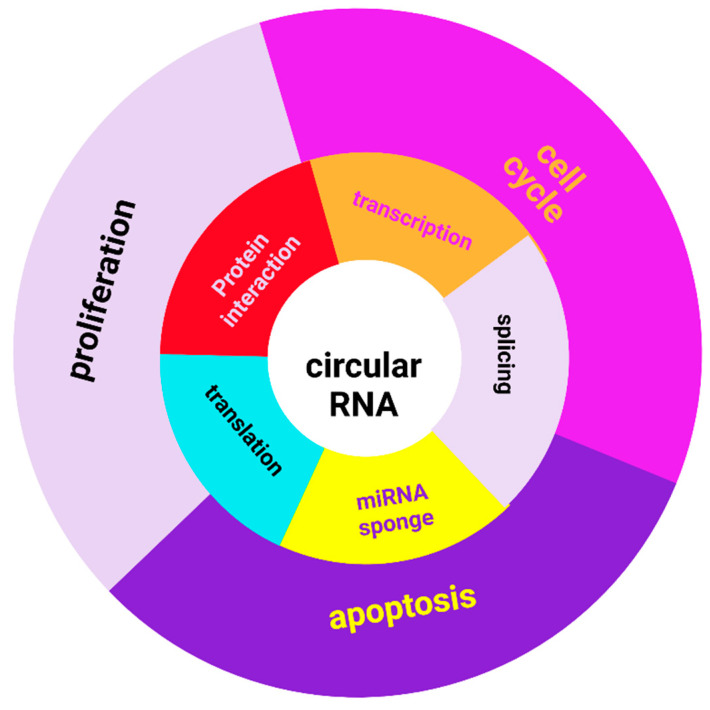
CircRNA functions and involvement in cellular processes. Created with BioRender.com.

**Table 1 biomolecules-11-01755-t001:** CircRNA in cancer [16].

Cancer Type	circRNAs (No)	circRNAs Downregulated (No)	circRNAs Upregulated (No)	circRNAs N/A
Acute Lymphoblastic Leukaemia	1	-	-	1
Acute myeloid leukaemia	11	4	5	2
Basal cell carcinoma	18	7	10	1
Bladder cancer	19	9	10	-
Breast cancer	82	18	28	36
Cervical cancer/carcinoma	18	-	17	1
Cholangiocarcinoma	1	1	-	-
Clear cell renal cell carcinoma	2	2	-	-
Kidney clear cell carcinoma	1	-	1	-
Colon cancer	6	5	-	1
Colorectal cancer	42	18	20	4
Cutaneous squamous cell carcinoma	5	2	3	-
Endometrial cancer	2	-	2	-
Epithelial Ovarian carcinoma	3	-	-	3
Oesophageal cancer	7	4	3	-
Oesophageal squamous cell carcinoma	34	12	22	-
Gastric cancer	78	41	27	10
Glioblastoma	46	19	8	3/16 *
Hepatoblastoma	17	9	8	-
Hepatocellular carcinoma	22	13	4	5
Hypopharyngeal squamous cell carcinoma	6	3	3	-
Laryngeal squamous cell cancer tissues	4	1	1	2
Liver cancer	5	-	3	2
Lung adenocarcinoma	4	1	3	-
Lung cancer	8	3	4	1
Non-small cell lung cancer	9	1	6	2
Malignant melanoma	1	-	-	1
Oral squamous cell carcinoma	9	1	1	7
Osteosarcoma	11	1	7	3
Pancreatic cancer	4	1	3	-
Pancreatic ductal adenocarcinoma	12	5	7	-
Papillary thyroid carcinoma	17	3	13	1
Prostate adenocarcinoma	1	1	-	-
Prostate cancer	1	-	1	-

N/A—not available; * unclear.

**Table 2 biomolecules-11-01755-t002:** CircRNAs in cancers.

Cancer Type	circRNAs	AUC	References
Acute Lymphoblastic Leukaemia	hsa_circ_0012152	0.8625	[64]
	hsa_circ_0001857	0.909	
Acute myeloid leukaemia	hsa_circ_0004277	0.957	[65]
Bladder cancer	hsa_circ_0018069	0.709	[66]
	circASXL1	0.77	
	hsa_circ_0077837	0.775	
	hsa_circ_0004826	0.79	
	circ0006332	0.86	
	circ_0137439	0.89	
Breast cancer	hsa_circ_103110	0.63	[67,68,69]
	hsa_circ_104689	0.61	
	hsa_circ_104821	0.60	
	hsa_circ_006054	0.71	
	hsa_circ_100219	0.78	
	hsa_circ_406697	0.64	
	hsa_circ_0001785	0.771	
	hsa_circ_0104824	0.823	
Cholangiocarcinoma	hsa_circ_0000673	0.85	[70,71]
	Cdr1as	0.740	
Clear cell renal cell carcinoma	circEHD2	0.757	[72]
	circNETO2	0.705	
	circEGLN3	0.879	
Colorectal cancer	circ_0001178	0.945	[73]
	circCDC66	0.884	
	circITGA7	0.879	
	circ_0000567	0.865	
	circ_0001649	0.857	
	circ_0003906	0.818	
	circ_0000826	0.816	
	circ_0000711	0.810	
	circ_001988	0.788	
Endometrial cancer	circ_0001776	0.7389	[74]
Epithelial Ovarian carcinoma	circBNC2	0.923	[75]
Oesophageal squamous cell cancer	hsa_circ_0001946	0.894	[76]
	hsa_circ_0062459	0.836	
Gastric cancer	hsa_circ_0007507	0.832	[77,78,79]
	hsa_circ_0000190	0.75	
	hsa_circ_0000096	0.82	
Glioblastoma	circFOXO3	0.870	[80,81]
	circ_0029426	0.730	
	circ-SHPRH	0.960	
	circHIPK3	0.855	
	circSMARCA5	0.823	
Hepatoblastoma	circHMGCS1	0.8971	[82]
Hepatocellular carcinoma	exo_circ_0006602	0.907	[79,83,84]
	circSMARCA5	0.938	
	hsa_circ_0005075	0.94	
Hypopharyngeal squamous cell carcinoma	circMORC3	0.767	[85]
Laryngeal squamous cell cancer tissues	hsa_circ_0036722	0.838	[85]
Liver cancer	circZFR	0.7069	[86]
	circFUT8	0.7575	
	circIPO11	0.7103	
Lung adenocarcinoma	hsa_circ_0005962	0.73	[87,88,89]
	hsa_circ_0086414	0.78	
	hsa_circ_0013958	0.815	
	hsa_circ_0000729	0.815	
Lung cancer	circ_102231	0.9	[90]
Non-small cell lung cancer	circFARSA	0.71	[90,91]
	circ_0079530	0.76	
	circRNA-FOXO3	0.78	
	circ_0014130	0.89	
	circ_0047921	0.757	
	circ_0056285	0.625	
	circ_0007761	0.750	
Oral squamous cell carcinoma	hsa_circ_0001874	0.922	[92]
	hsa_circ_0001971	0.922	
	circ-MMP9	0.91	
	circMAN1A2	0.799	
	circSPATA6	0.7748	
Osteosarcoma	circ_0000190	0.889	[93]
	circ_0000885	0.783	
	circ_HIPK3	0.783	
	circPVT1	0.871	
	circ_0081001	0.898	
Pancreatic cancer	circ-LDLRAD3	0.67	[94,95]
	hsa_circ_0013587	0.6995	

AUC—Area Under the Curve.

## Data Availability

Some data were obtained from circAtlas2.0. Available online: http://circatlas.biols.ac.cn/ (accessed on 25 October 2021).

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
