# Peer review of "Circular RNA—Is the Circle Perfect?"

_biomolecules, 2021, doi:10.3390/biom11121755_

Round 1

Reviewer 1 Report

The review manuscript “Circular RNA – is the circle perfect? ” by Caba et al., summarized diverse aspects of circular RNAs in recent findings. This review could provide circular RNA as potential biomarkers into various cancer types. Overall the manuscript was well written and cited important publications in this field. However, I listed major comments as followed.

Page 3 line 107 - In figure 1, there are many abbreviates such as sd, sa, RNAB, authors should describe what are those. In addition, what are red bars between exons? If those are ALU repeats then describe them in the figure caption. Diagram for the backspacing is not clear to show ericRNA and EIciRNA.

Page4 line 135 -  2.5 Regulation looks like part of 2.4 Biogenesis and these are still unclear to describe separate section.

Page4 line 149 -  In Translation author described more m6A modification than translation. m6A and other RNA modifications should be an independent section to describe cirRNA function. Furthermore, it would be better to add an independent section with numbers in CirRNA-protein interactions and Binding or sequestration of proteins

Page 7 and 8 - There are so many AUC values in all the cirRNAs. AUC just reflects how their model worked well. The author simply adds more cirRNAs in Table 2.

Page 10 - Why AUC is a separate section? The paragraph is not matched to the title. If AUC is the part of Table 2 then remove bold. 

Page 10 - The author emphasizes the next step of circRNAs to 5P medicine in conclusion. The author should give any examples or ideas on how circRNAs be served as personalized, preventive, predictive, participatory, and precision.

Author Response

“The review manuscript “Circular RNA – is the circle perfect? ” by Caba et al., summarized diverse aspects of circular RNAs in recent findings. This review could provide circular RNA as potential biomarkers into various cancer types. Overall the manuscript was well written and cited important publications in this field. However, I listed major comments as followed.”

Answer: Thank you very much for your comments and constructive observations.

“Page 3 line 107 - In figure 1, there are many abbreviates such as sd, sa, RNAB, authors should describe what are those. In addition, what are red bars between exons? If those are ALU repeats then describe them in the figure caption. Diagram for the backspacing is not clear to show ericRNA and EIciRNA.”

Answer: We redid the figure and added the explanations for the abbreviations used to the legend

“Page4 line 135 -  2.5 Regulation looks like part of 2.4 Biogenesis and these are still unclear to describe separate section.”

Answer: We gave up section 2.5 and the information is framed in 2.4.

“Page4 line 149 -  In Translation author described more m6A modification than translation. m6A and other RNA modifications should be an independent section to describe cirRNA function. Furthermore, it would be better to add an independent section with numbers in CirRNA-protein interactions and Binding or sequestration of proteins”

Answer: We created independent sections for 2.5.1. m6A modifications, 2.5.2. CircRNA-proteins interactions, 2.5.3. Binding or sequestration of proteins

“Page 7 and 8 - There are so many AUC values in all the cirRNAs. AUC just reflects how their model worked well. The author simply adds more cirRNAs in Table 2.”

Answer: We added more circRNAs in the Table. In the section after the table we made comparisons between the efficiency of use of classic serum markers / circRNA + classic serum markers / combinations of circRNAs.

“Page 10 - Why AUC is a separate section? The paragraph is not matched to the title. If AUC is the part of Table 2 then remove bold.”

Answer: AUC is not a separate section. It is a part of Table 2. We removed bold. 

“Page 10 - The author emphasizes the next step of circRNAs to 5P medicine in conclusion. The author should give any examples or ideas on how circRNAs be served as personalized, preventive, predictive, participatory, and precision.”

Answer: We rewrote the conclusions.

The manuscript was verified for language mistakes and rectified accordingly.

Reviewer 2 Report

Major comments:

  1. The last subsection of "section-3. Circular RNA- potential biomarker" start with "AUC- Area Under the Curve 
    Zhao et al demonstrated the existence of a set of circRNAs involved in the pathogenesis of Moyamoya disease by modulating MAPK signalling"  is not clear as what does it mean by the heading AUC?
  2. The section-3 states the role of circular RNA as potential biomarker (not specifically mention cancer) but it mainly highlights in various types of carcinomas (in addition to Moyamoya disease), so its better to include the examples of CiRNA biomarkers for other human diseases too (like diabetes, tuberculosis....) or change the heading to "circular RNA as potential biomarker in cancer" and remove the section for Moyamoya disease.    

Minor comments: (Reframe the sentences/check for English)

A) Most circRNA molecules contain exons from genes encoding proteins, but
they provide also from introns, intergenic regions, UTRs, ncRNA loci and antisense locations of known transcripts Other newly described categories are added to these: fusion circRNA (f-circRNAs), read-through circRNAs (rt-circRNAs) and mitochondria-encoded circRNA (mecciRNAs). 

B) Stable circRNAs was discovered in exosomes where intervene in transcriptional and translational regulation, splicing regulation, miRNA sponge and protein inhibition [27]. 

C) For other cancers, potential circRNA diagnostic biomarkers were synthesized (hint: summarized) in Table 2.

Author Response

Thank you very much for your comments and constructive observations.

Major comments:

  1. “The last subsection of "section-3. Circular RNA- potential biomarker" start with "AUC- Area Under the Curve 
    Zhao et al demonstrated the existence of a set of circRNAs involved in the pathogenesis of Moyamoya disease by modulating MAPK signalling"  is not clear as what does it mean by the heading AUC?”

AUC is not a separate section. It is a part of Table 2. We remove bold. 

  1. “The section-3 states the role of circular RNA as potential biomarker (not specifically mention cancer) but it mainly highlights in various types of carcinomas (in addition to Moyamoya disease), so its better to include the examples of CiRNA biomarkers for other human diseases too (like diabetes, tuberculosis....) or change the heading to "circular RNA as potential biomarker in cancer" and remove the section for Moyamoya disease”.    

Answer: We changed the heading to "circular RNA - a potential biomarker in cancer" and we removed the section for Moyamoya disease.    

“Minor comments: (Reframe the sentences/check for English)

  1. Most circRNA molecules contain exons from genes encoding proteins, but
    they provide also from introns, intergenic regions, UTRs, ncRNA loci and antisense locations of known transcripts Other newly described categories are added to these: fusion circRNA (f-circRNAs), read-through circRNAs (rt-circRNAs) and mitochondria-encoded circRNA (mecciRNAs).” 

Answer: We rewrote the phrase: “Most circRNA molecules contain exons from genes encoding proteins, but they originate also from introns, intergenic regions, UTRs, ncRNA loci and antisense locations of known transcripts. Others newly described categories are added: fusion circRNA (f-circRNAs), read-through circRNAs (rt-circRNAs) and mitochondria-encoded circRNA (mecciRNAs).”

  1. “Stable circRNAs was discovered in exosomes where intervene in transcriptional and translational regulation, splicing regulation, miRNA sponge and protein inhibition [27].” 

Answer: We rewrote the phrase “Stable circRNAs was discovered in exosomes with a role in the transcriptional and translational regulation, splicing regulation, miRNA sponge and protein inhibition [27].”

  1. “For other cancers, potential circRNA diagnostic biomarkers were synthesized (hint: summarized)in Table 2.”

We modified it.

The manuscript was verified for language mistakes and rectified accordingly.

Round 2

Reviewer 1 Report

The manuscript “Circular RNA – is the circle perfect? ” by Caba et al., addressed all the comments from reviewers. Tables and contents are improved overall. I don’t have any further comments. Thank you.